# Restoration of the healing microenvironment in diabetic wounds with matrix-binding IL-1 receptor antagonist

Jean L. Tan[1,3], Blake Lash [1,2,3], Rezvan Karami[1], Bhavana Nayer [1], Yen-Zhen Lu[1], Celeste Piotto[1], Ziad Julier[1] & Mikaël M. Martino [1✉]

Chronic wounds are a major clinical problem where wound closure is prevented by pathologic factors, including immune dysregulation. To design efficient immunotherapies, an understanding of the key molecular pathways by which immunity impairs wound healing is needed. Interleukin-1 (IL-1) plays a central role in regulating the immune response to tissue injury through IL-1 receptor (IL-1R1). Generating a knockout mouse model, we demonstrate that the IL-1–IL-1R1 axis delays wound closure in diabetic conditions. We used a protein engineering approach to deliver IL-1 receptor antagonist (IL-1Ra) in a localised and sustained manner through binding extracellular matrix components. We demonstrate that matrix-binding IL-1Ra improves wound healing in diabetic mice by re-establishing a pro-healing microenvironment characterised by lower levels of pro-inflammatory cells, cytokines and senescent fibroblasts, and higher levels of anti-inflammatory cytokines and growth factors. Engineered IL-1Ra has translational potential for chronic wounds and other inflammatory conditions where IL-1R1 signalling should be dampened.

[1] European Molecular Biology Laboratory Australia, Australian Regenerative Medicine Institute, Monash University, Melbourne, VIC, Australia. [2] Department of Biological Engineering, Massachusetts Institute of Technology, Cambridge, MA, USA. [3] These authors contributed equally: Jean L. Tan, Blake Lash.
✉email: mikael.martino@monash.edu

Chronic wounds have become a major challenge to healthcare systems worldwide, potentially affecting a number of at-risk populations including diabetic and elderly patients, and those that remain bedridden[1]. While the causes leading to impaired wound healing are diverse, chronic wounds have many common features including excessive levels of pro-inflammatory immune cells and cytokines, high concentration of proteases, low levels of growth factors and a higher number of senescent cells[1,2]. Because of these cellular and molecular characteristics, chronic wounds are usually trapped in the inflammation phase of the healing process and fail to progress towards the later phases of healing in an orderly and timely manner[2]. The persistent inflammation preventing the progression of chronic wounds to an anti-inflammatory/repair state is likely due to dysregulation of immune signalling. Indeed, inadequate immune responses and imbalance in immune signalling are common in diabetic and elderly patients, where high levels of the pro-inflammatory cytokine interleukin-1 (IL-1) play a central role[3]. For example, obesity and hyperglycaemia are known to induce expression of IL-1β in a number of different cell types, including immune cells such as monocytes/macrophages[4–7].

The impact of IL-1β in chronic wounds is still elusive. However, IL-1β is known to act as an upstream signal for sustaining inflammasome activity in wound macrophages, in addition to inducing inflammatory signals in other cell types. Since activation of the inflammasome further promotes the release of IL-1β, it could be part of an inflammatory positive-feedback loop that prevents polarisation of macrophages towards an anti-inflammatory phenotype[8]. More generally, an excess of IL-1β–driven inflammatory signals in wounds may trigger a cascade of events that delay wound closure. For instance, these events could include slow clearance of inflammatory cells from the wound, senescence of fibroblasts and degradation of pro-repair growth factors and extracellular matrix (ECM) proteins due to high levels of matrix metalloproteinases (MMPs). Therefore, blocking IL-1β or its signalling through IL-1 receptor 1 (IL-1R1) may be a promising option to reduce the persistent inflammation in chronic wounds and to promote healing[9,10]. Studies have shown that recombinant IL-1 receptor antagonist (IL-1Ra) effectively blocks IL-1R1 signalling when delivered in 100 to 1000-fold excess[11], however, given the short half-life of the protein in vivo, this is difficult to practically sustain[12]. The use of very high doses of inhibitors/antagonists for IL-1R1 may circumvent this issue, but the approach may also trigger systemic adverse effects such as immunogenicity and opportunistic infections[13,14]. Therefore, the use of much lower doses is preferable. Yet, blocking IL-1R1 signalling efficiently and locally with low doses of therapeutics is challenging and would require advanced drug delivery systems.

In this study, we first generated a knockout mouse model to evaluate the impact of IL-1R1 in chronic wounds. Then, we thought of utilising a protein engineering approach to locally block IL-1R1 signalling by delivering a modified form of IL-1Ra which bears super-affinity to the ECM. The system allows the engineered therapeutic factor to be effective at a very low dose without the need of a biomaterial carrier[15]. We also decided to compare this approach to a growth factor-based strategy, by utilising platelet-derived growth factor-BB (PDGF-BB) which is currently approved for the treatment of chronic wounds[16].

## Results

### IL-1R1 signalling delays wound healing in diabetic mice.
IL-1R1 signalling in tissues is tightly regulated by the ratio between IL-1β and IL-1Ra[17,18]. Thus, we first tested if the levels of IL-1β and IL-1Ra differ in skin wounds that display normal or impaired healing. As a model of impaired healing, we chose full-thickness wounds in diabetic mice ($Lepr^{db/db}$), since it is a widely accepted and relevant experimental model that mimics some aspects of chronic wounds in humans[19,20]. We measured the concentrations of IL-1β and IL-1Ra in wounds of $Lepr^{db/db}$ (diabetic) and of $Lepr^{+/db}$ (non-diabetic littermates) mice at various time points post injury. At every time point in diabetic mice, we found significantly elevated levels of IL-1β and significantly lower levels of IL-1Ra, compared to non-diabetic mice (Fig. 1a). Because macrophages are known to be a significant source of IL-1β[21], we compared IL-1β secretion by $Lepr^{db/db}$ and $Lepr^{db/+}$ macrophages. Interestingly, $Lepr^{db/db}$ macrophages secreted significantly more IL-1β in a steady-state, as well as upon stimulation with lipopolysaccharide (LPS) and adenosine triphosphate (ATP) (Fig. 1b). Then, in order to evaluate the actual impact of IL-1β and IL-1R1 signalling in wounds that present an impaired healing, we crossed $Lepr^{db/db}$ mice with $Il1r1^{-/-}$ mice to obtain diabetic mice deficient for IL-1R1 ($Lepr^{db/db}$-$Il1r1^{-/-}$) (Fig. 1c and Supplementary Fig. 1). Remarkably, diabetic mice deficient in IL-1R1 healed wounds much faster with near-complete re-epithelization after 9 days, while diabetic wounds with IL-1R1 signalling were ~50% open (Fig. 1d, e).

### IL-1Ra fused to PlGF$_{123–141}$ is extensively retained in skin tissue.
Because wounds of diabetic mice deficient for IL-1R1 showed much faster healing, we hypothesised that delivery of IL-1Ra into chronic wounds would promote faster closure. We chose to compare this strategy to the delivery of PDGF-BB, since this growth factor is well recognised to enhance wound healing and could serve as a clinically relevant positive control[16]. Yet, promoting tissue repair with local delivery of recombinant cytokines or growth factors usually requires the use of a delivery system that ideally allows retention of low doses of the therapeutic at the delivery site and limits its diffusion to prevent side effects. A local controlled release of therapeutics may be achieved with biomaterial systems. However, instead of utilising a biomaterial-based delivery strategy for IL-1Ra and PDGF-BB, we opted for a biomaterial-free option. Here, we enhanced the affinity of the recombinant proteins for endogenous ECM components by fusing them to an ECM-binding sequence derived from the heparin-binding domain of placenta growth factor (PlGF$_{123–152}$)[15]. First, we determined the smallest ECM-binding sequence derived from PlGF$_{123–152}$ by producing seven truncated versions of PlGF$_{123–152}$ and testing their binding to common ECM proteins (fibronectin, vitronectin, tenascin C and fibrinogen) and heparan sulfate. We found that a shorter version of the ECM-binding sequence, PlGF$_{123–141}$, strongly binds all ECM proteins tested, as well as heparan sulfate (Fig. 2a, b). Then, we recombinantly fused PlGF$_{123–141}$ at the C terminus of IL-1Ra and PDGF-BB to generate IL-1Ra/PlGF$_{123–141}$ and PDGF-BB/PlGF$_{123–141}$ (Fig. 2c). The engineered factors displayed a great increase in affinity for the ECM protein tested (Supplementary Fig. 2, Supplementary Table 1)[22].

We then verified that fusing PlGF$_{123–141}$ to IL-1Ra and PDGF-BB did not compromise their activity, by utilising cell types known to respond to these factors. Stimulation of dermal fibroblast proliferation was chosen to verify activity of engineered PDGF-BB, while the activity of engineered IL-1Ra was verified by measuring its capacity to inhibit IL-6 release by macrophages in response to IL-1β. The proliferation of dermal fibroblasts following stimulation with wild-type PDGF-BB and PDGF-BB/PlGF$_{123–141}$ was similar, and the inhibitory activities of IL-1Ra and IL-1Ra/PlGF$_{123–141}$ on IL-1β–stimulated macrophages were comparable (Supplementary Fig. 3). These results indicated that fusion to PlGF$_{123–141}$ does not alter the activity of the engineered factors.

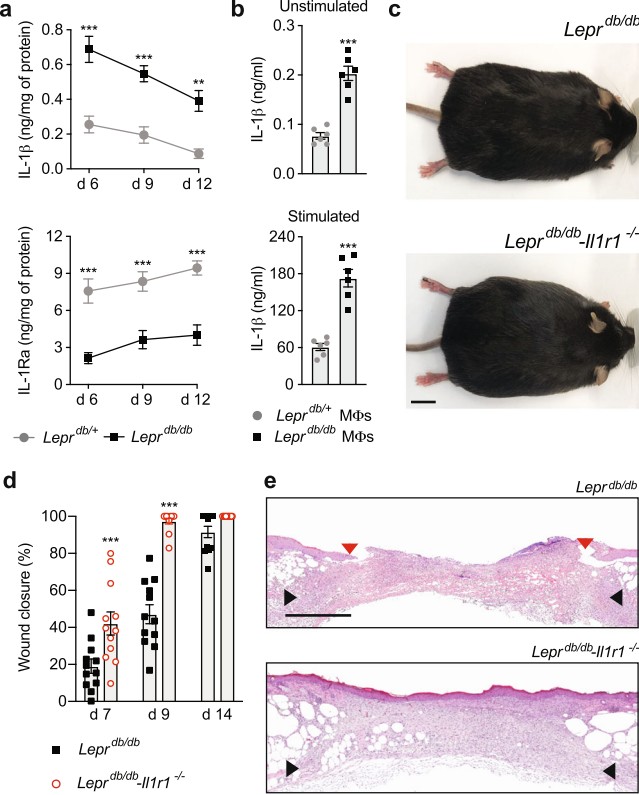

**Fig. 1 IL-1R1 signalling delays wound healing in diabetic mice. a** Full-thickness wounds (5 mm) were created in diabetic mice (Lepr$^{db/db}$) and non-diabetic littermates (Lepr$^{db/+}$). Concentrations of IL-1β and IL-1Ra in wounds harvested at various time points. $n = 4$ wounds per time point. **b** Concentrations of IL-1β detected in media of unstimulated or stimulated (LPS + ATP) bone marrow-derived macrophages (MΦs) from Lepr$^{db/+}$ and Lepr$^{db/db}$ mice. **c** Lepr$^{db/db}$ mice were crossed with Il1r1$^{-/-}$ mice to generate diabetic mice deficient for IL-1R1. Representative pictures of 14-week-old mice are shown. Scale bar = 1 cm. **d, e** Full-thickness wounds were created in Lepr$^{db/db}$ and Lepr$^{db/db}$-Il1r1$^{-/-}$ mice. Graph in **d** shows wound closure kinetics evaluated by histomorphometric analysis of tissue sections. $n = 12$ wounds per time point. Representative histology (haematoxylin and eosin staining) after 9 days are shown in (**e**). Black arrows indicate wound edges and red arrows indicate tips of epithelium tongue. The epithelium (if any) appears in purple as a homogeneous keratinocyte layer on top of the wounds. The granulation tissue under the epithelium contains granulocytes with dark-purple nuclei. Fat tissue appears as transparent bubbles. Scale bar = 1 mm. For (**a**), (**b**) and (**c**), data are means ± SEM. For (**a**) and (**d**), two-way ANOVA with Bonferroni *post hoc* test for pair-wise comparisons. For (**b**), two-tailed Student's *t* test. **$P \leq 0.01$, ***$P \leq 0.001$.

Next, we assessed the capacity of IL-1Ra and PDGF-BB to bind the ECM, utilising an ECM-mimetic hydrogel composed of common ECM components. The ECM-mimetic hydrogel is based on a fibrin matrix functionalised with fibronectin, vitronectin, tenascin C and heparan sulfate. All ECM components are naturally incorporated in the hydrogel thanks to protein–protein interactions, following thrombin-induced fibrin polymerisation. Fibronectin is crosslinked to fibrin via the transglutaminase factor XIIIa (present in fibrinogen preparations)[23]. Vitronectin directly interacts with fibrin[24], while tenascin C binds fibronectin[25]. Fibrin, fibronectin, vitronectin and tenascin C all have heparin-binding domains capturing heparan sulfate in the hydrogel (Fig. 2d). Both IL-1Ra/PlGF$_{123-141}$ and PDGF-BB/PlGF$_{123-141}$ were retained in the ECM-mimetic hydrogel while the wild-type forms were quickly released. Moreover, IL-1Ra/PlGF$_{123-141}$ and

PDGF-BB/PlGF$_{123-141}$ were gradually released in the presence of the protease plasmin which cleaves PlGF$_{123-141}$, as well as the ECM proteins that form the hydrogel[15] (Fig. 2e). To confirm this strong ECM-binding activity in vivo, we then tested the retention of the engineered factors after intradermal administration. As anticipated, IL-1Ra and PDGF-BB fused to PlGF$_{123-141}$ showed extended retention in tissue with about 30–40% of the initial dose retained after 5 days (Fig. 2f).

**Topical delivery of super-affinity IL-1Ra accelerates wound healing in diabetic mice.** To evaluate whether localised delivery of IL-1Ra variants promotes healing of diabetic wounds, we utilised the Lepr$^{db/db}$ mouse again. A single low dose of IL-1Ra or PDGF-BB variants were delivered topically in saline following full-thickness wounding (0.5 μg of IL-1Ra and PDGF-BB, and equimolar doses of the engineered forms). One week after treatment, wounds that received IL-1Ra/PlGF$_{123-141}$ showed significantly more closure—characterised by the extent of re-epithelization—compared to saline control, IL-1Ra and both forms of PDGF-BB (Fig. 3a, b and Supplementary Fig. 4a, b). Near 100% closure was observed 9 days post treatment in wounds that received IL-1Ra/PlGF$_{123-141}$ or PDGF-BB/PlGF$_{123-141}$, while wounds treated with saline, IL-1Ra or PDGF-BB were still largely open (Fig. 3a, b and Supplementary Fig. 4a, b). To verify that the PlGF$_{123-141}$ sequence itself has no influence on wound healing, we tested the effect of treating wounds with PlGF$_{123-141}$ fused to the non-relevant protein GST. As expected, delivering GST/PlGF$_{123-141}$ had no significant effect on wound healing (Supplementary Fig. 4c, d). Because angiogenesis is critical for wound healing, we looked at the extent of angiogenesis at 9 days post treatment by immunostaining for endothelial and smooth muscle cells (CD31 and desmin, respectively). Wound treatment with IL-1Ra/PlGF$_{123-141}$ and PDGF-BB/PlGF$_{123-141}$ induced greater angiogenesis compared to the wild-type factors (Fig. 3c, d and Supplementary Fig. 5a). Then, we assessed the epithelial barrier properties of IL-1Ra/PlGF$_{123-141}$-treated wounds 9 days post treatment, by measuring their surface electrical capacitance[26,27]. Wounds treated with saline showed high capacitance values, while wounds treated with IL-1Ra/PlGF$_{123-141}$ showed values that were similar to those measured on uninjured skin, indicating low fluid leakage and, therefore, reformation of an epithelial barrier (Supplementary Fig. 5b). Finally, we tested whether IL-1Ra/PlGF$_{123-141}$ has an effect on non-diabetic mice. Compared with saline control, treatment with IL-1Ra/PlGF$_{123-141}$ led to a significant but modest improvement of wound closure in wild-type mice (Supplementary Fig. 6a, b), suggesting that blocking IL-1R1 signalling in animals without immune dysregulation has likely less impact on wound healing.

Based on these results, we hypothesised that treatment with IL-1Ra would lead to a series of molecular events resulting from the inhibition of IL-1R1 signalling (Fig. 3e). IL-1β is known to trigger a strong inflammatory response that induces the mobilisation of inflammatory neutrophils and macrophages[17]. Therefore, we tested to which extent IL-1Ra/PlGF$_{123-141}$ modulates neutrophil and macrophage accumulation in wounds during the healing process, using flow cytometry (Supplementary Fig. 7). Interestingly, treatment of wounds with IL-1Ra/PlGF$_{123-141}$ promoted faster clearance of neutrophils (CD11b$^+$, Ly6G$^+$ cells) and accumulation of more macrophages (F4/80$^+$, CD11b$^+$ cells), compared to treatment with wild-type IL-1Ra or saline (Fig. 3f). Moreover, the expression levels of CD206—a marker for M2-like macrophages—were significantly higher in the IL-1Ra/PlGF$_{123-141}$-treated group, indicating that wound macrophages were likely more anti-inflammatory (Fig. 3f).

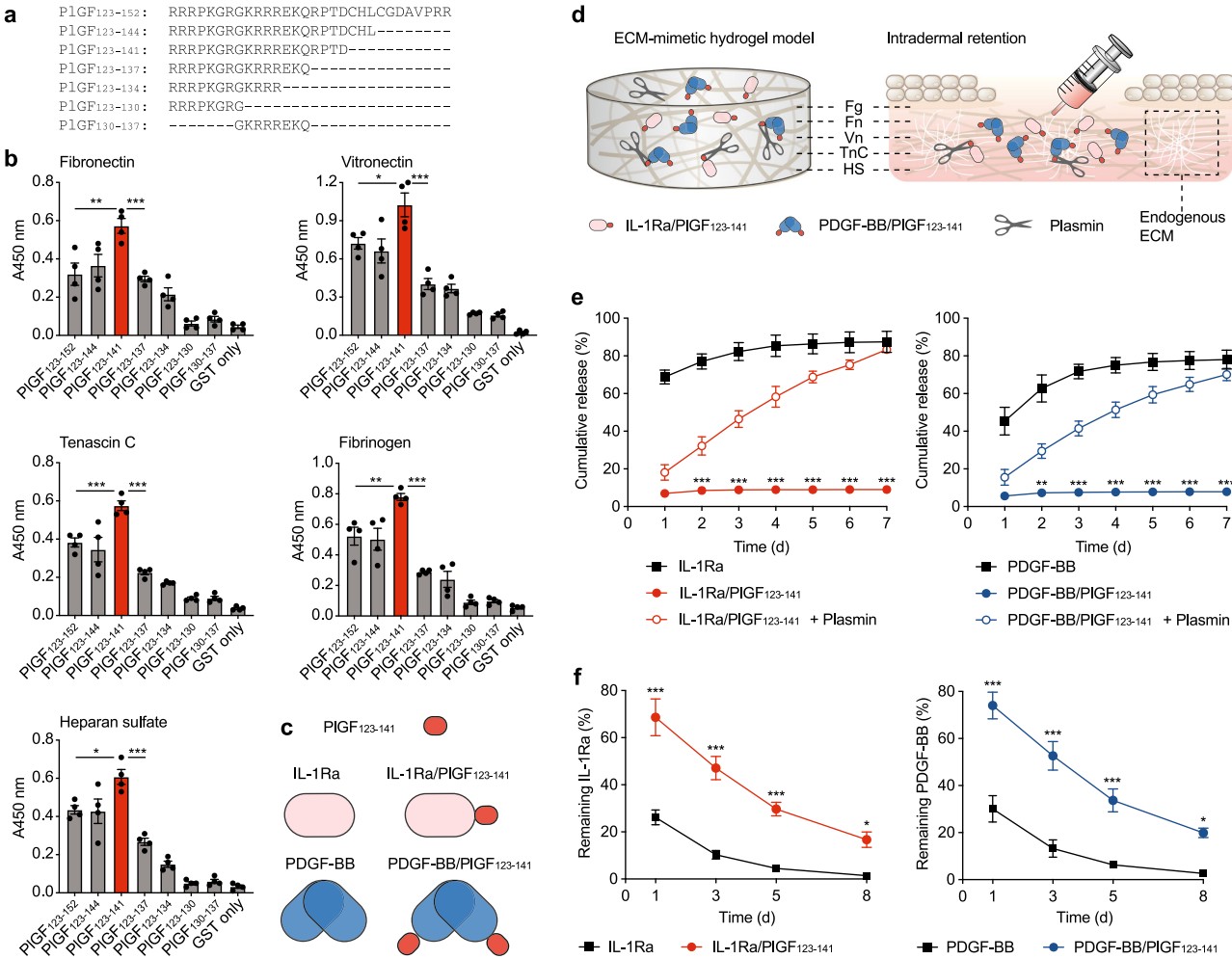

**Fig. 2 PlGF₁₂₃₋₁₄₁-fused IL-1Ra and PDGF-BB strongly bind ECM and are retained in skin tissue. a** Amino-acid sequences of PlGF fragments fused to glutathione S-transferase (GST). **b** ELISA plates were coated with ECM proteins and incubated with PlGF fragments. Graphs show signals given when detecting GST. PlGF₁₂₃₋₁₄₁ is shown in red. $n = 4$. **c** PlGF₁₂₃₋₁₄₁ (in red) was added to the C terminus of IL-1Ra (in pink) or PDGF-BB (in blue) to generate IL-1Ra/PlGF₁₂₃₋₁₄₁ and PDGF-BB/PlGF₁₂₃₋₁₄₁. PDGF-BB occurs as a dimer. **d** Schematic representation of the ECM-mimetic hydrogel and skin endogenous ECM. Fg fibrinogen, Fn fibronectin, Vn vitronectin, TnC tenascin C, HS heparan sulfate. **e** ECM-mimetic hydrogels were generated with IL-1Ra or PDGF-BB variants and incubated in ten times volume of buffer (with or without plasmin) that was changed every 24 h. The graphs show the cumulative release of IL-1Ra or PDGF-BB variants in buffer. $n = 4$. **f** The percentage of IL-1Ra and PDGF-BB variants remaining in skin after intradermal injection was measured at various time points. $n = 4$ per time point. For (**b**), (**e**) and (**f**), data are means ± SEM. For (**b**), one-way ANOVA with Bonferroni *post hoc* test for pair-wise comparisons. For (**e**) and (**f**), two-way ANOVA with Bonferroni *post hoc* test for pair-wise comparisons. $*P ≤ 0.05$, $**P ≤ 0.01$, $***P ≤ 0.001$.

**Treatment with super-affinity IL-1Ra promotes a pro-healing microenvironment.** Levels of cytokines, MMPs and growth factors in the wound microenvironment are critical for the progression of the healing process. Thus, we investigated how treatment with super-affinity IL-1Ra influences the wound levels of these factors at different time points post treatment (Fig. 4a and Supplementary Fig. 8). We first measured levels of the pro-inflammatory cytokines IL-1β, IL-6 and CXC chemokine ligand (CXCL) 1 (a neutrophil chemoattractant), as well as levels of the anti-inflammatory cytokines transforming growth factor-β1 (TGF-β1), IL-4 and IL-10. Remarkably, compared to treatment with saline and IL-1Ra, IL-1Ra/PlGF₁₂₃₋₁₄₁ significantly reduced the pro-inflammatory factors and increased the anti-inflammatory factors. Next, we measured the levels of MMP-2 and MMP-9 which are typically found in high concentrations in chronic wounds and are known to degrade ECM components and growth factors[16,28–31]. Delivering IL-1Ra/PlGF₁₂₃₋₁₄₁ significantly decreased MMP-2 and MMP-9 but increased the tissue inhibitor of metallopeptidase-1 (TIMP-1). Finally, we focused on fibroblast

growth factor-2 (FGF-2), PDGF-BB and vascular endothelial growth factor-A (VEGF-A) which are key growth factors secreted by macrophages and other cells during wound healing[32,33]. Delivering IL-1Ra/PlGF₁₂₃₋₁₄₁ significantly enhanced the levels of these pro-healing factors compared to saline and IL-1Ra.

Since fibroblast senescence is a major hurdle in diabetic wound healing[34–37], we explored the effect of IL-1Ra treatment on wound fibroblast senescence. First, we tested the effect of IL-1β on dermal fibroblasts in vitro, by measuring senescence-associated β-galactosidase (SA-β-gal), as increased SA-β-gal is considered a hallmark for senescence[38]. Interestingly, we found that treatment of primary dermal fibroblasts with IL-1β increases SA-β-gal activity, suggesting that the cytokine accelerates cellular senescence (Fig. 4b). We also showed that stimulation of fibroblasts with IL-1β induces the secretion of typical senescence-associated factors including IL-6, CC chemokine ligands 1 (CCL1) and 3 (CCL3), CXCL1, 2 and 10, as well as MMP-2 (Supplementary Fig. 9). To confirm this phenomenon in vivo, we tested whether treatment of diabetic wounds with

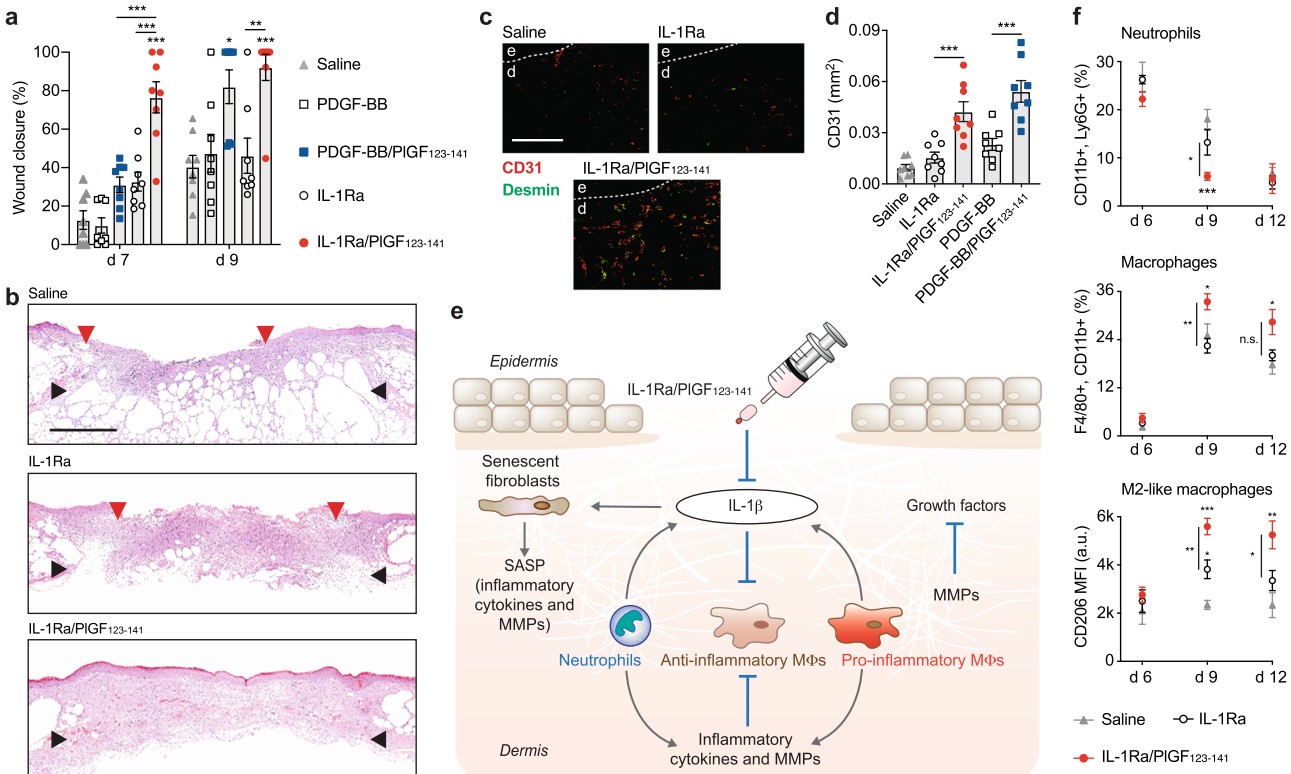

**Fig. 3 Super-affinity IL-1Ra accelerates wound healing in diabetic mice. a, b** Full-thickness wounds in *Lepr*^db/db^ were treated with IL-1Ra or PDGF-BB variants (0.5 µg of wild types, equimolar of engineered versions). Wound closure evaluated by histomorphometric analysis of tissue sections in (**a**). n = 8 wounds per condition. Representative histology (haematoxylin and eosin staining) 9 days post treatment shown in (**b**). Black arrows indicate wound edges and red arrows indicate tips of epithelium tongue. The epithelium (if any) appears in purple as a homogeneous keratinocyte layer on top of the wounds. The granulation tissue under the epithelium contains granulocytes with dark-purple nuclei. Fat tissue appears as transparent bubbles. Scale bar = 1 mm. **c, d** Angiogenesis at day 9 assessed by immunostaining of wound sections for CD31 (endothelial cells, red) and desmin (smooth muscle cells, green). Representative images in (**c**). Dashed lines indicate separation between epidermis (indicated as **e**) and dermis (indicated as **d**). Scale bar = 0.2 mm. Quantification of CD31 in (**d**). n = 8. **e** Schematic representation of hypothesised IL-1Ra/PlGF₁₂₃₋₁₄₁ effects in diabetic wounds. MΦs macrophages, MMPs matrix metallopeptidases, SASP senescence-associated secretory phenotype; grey arrows represent induction; blue lines represent inhibition. **f** Neutrophil and macrophage populations in wounds measured by flow cytometry at various time points post wounding. Percentages were calculated over total live wound cells. Median fluorescence intensity (MFI) for CD206 was measured in macrophages (F4/80⁺, CD11b⁺ cells). n = 8 wounds. In (**a**), (**d**) and (**f**), data are means ± SEM. One-way ANOVA with Bonferroni *post hoc* test for pair-wise comparisons (significances shown are between saline and the other groups, unless indicated otherwise). *P ≤ 0.05, **P ≤ 0.001, ***P ≤ 0.001. ns non-significant.

IL-1Ra was able to reduce wound fibroblast senescence at day 9 post treatment, which was the time point at which we had observed almost complete wound closure. We found that both IL-1Ra variants, but IL-1Ra/PlGF₁₂₃₋₁₄₁ to a greater extent, were capable of decreasing SA-β-gal activity in fibroblasts. Fibroblasts in diabetic wounds 9 days post treatment with IL-1Ra/PlGF₁₂₃₋₁₄₁, displayed a level of SA-β-gal activity similar to fibroblasts in uninjured skin (Fig. 4c and Supplementary Fig. 10).

## Discussion

There has been a recent growing interest in developing therapies based on immunomodulation for chronic wounds and more generally for regenerative medicine[2]. For instance, a number of strategies have explored the use of biomaterials, recombinant cytokines, microRNAs and extracellular vesicles. However, translating these approaches into the clinic may be challenging, due to issues related to safety, scalability and cost-effectiveness[2]. In addition, a better understanding of the most important molecular pathways through which the immune system prevents or promotes the wound healing process is essential to design novel and effective therapies. Here, we theorised that IL-1R1 signalling via IL-1β is one of the most important pathways

preventing the maturation of a pro-repair microenvironment in chronic wounds.

IL-1R1 signalling is tightly controlled by IL-1Ra[17], but immune dysregulation may result in a high ratio of IL-1β to IL-1Ra leading to robust IL-1R1 signalling. We found that the IL-1β:IL-1Ra ratio is continuously higher in *Lepr*^db/db^ wounds, indicating that IL-1R1 signalling is probably much more pronounced in this mouse model that displays impaired wound healing. Interestingly, a similarly high ratio has been reported at the mRNA level in corneal wounds of streptozotocin-induced diabetic mice[9]. Further supporting that *Lepr*^db/db^ mice have higher IL-1R1 signalling levels, we found that IL-1β secretion is enhanced in *Lepr*^db/db^ macrophages derived from bone marrow. Our reasoning for using macrophages differentiated from bone marrow isolates comes from the fact that most wound macrophages originate from extravasating inflammatory blood monocytes (Ly6c^high^) that are recruited from the bone marrow following inflammatory signals[39,40]. Although the molecular mechanisms leading to higher secretion of IL-1β by *Lepr*^db/db^ macrophages are still elusive, higher expression of IL-1β in various cell types has been linked to obesity and hyperglycaemia[4–7].

The *Lepr*^db/db^ mouse is widely used in wound healing research, because the model mimics some aspects of human chronic

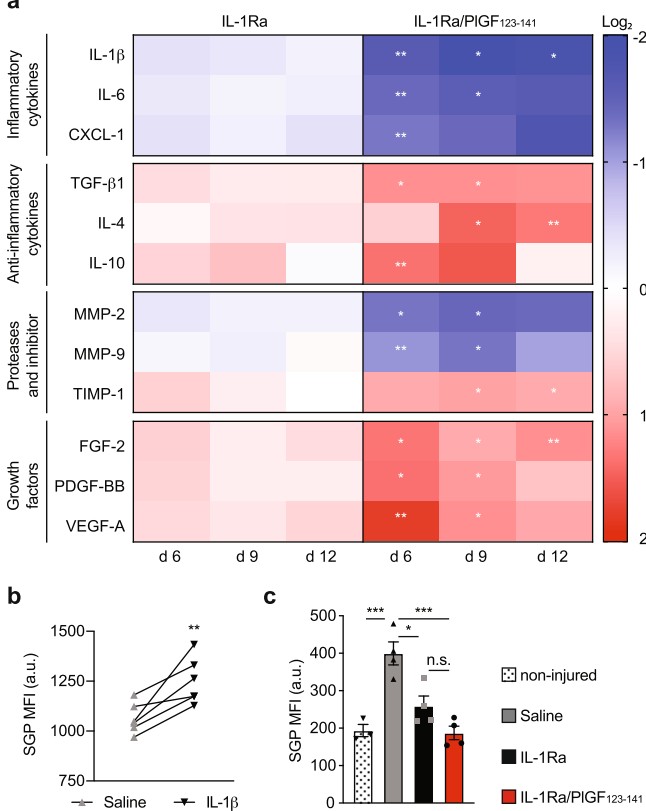

**Fig. 4 Super-affinity IL-1Ra leads to a pro-healing microenvironment.**
**a** Full-thickness wounds were created in *Lepr^db/db* mice and treated with saline control or IL-1Ra variants (0.5 µg of wild type, equimolar IL-1Ra/PlGF₁₂₃₋₁₄₁). After 6, 9 and 12 days wound concentrations of cytokines, MMPs, TIMP-1 and growth factors were measured by ELISA. The heat map shows fold change in log₂ over treatment with saline. n = 4. **b** Dermal fibroblasts were cultured with IL-1β (1 ng/ml) or saline control. After 9 days, SA-β-gal activity was assessed using senescence green probe (SGP). Graph shows median fluorescence intensity (MFI). n = 6. **c** Full-thickness wounds in *Lepr^db/db* mice were treated with saline or IL-1Ra variants. After 9 days, SA-β-gal activity in wound fibroblast was assessed by flow cytometry using SGP MFI. SA-β-gal activities were compared to fibroblasts in non-injured skin samples. n = 4. For all panels, data are means ± SEM. For (**a**), two-way ANOVA with Bonferroni *post hoc* test for pair-wise comparisons between IL-1Ra and IL-1Ra/PlGF₁₂₃₋₁₄₁. For (**b**), two-tailed Student's *t* test. For (**c**), one-way ANOVA with Bonferroni *post hoc* test for pair-wise comparisons. * P ≤ 0.05, ** P ≤ 0.01, ***P ≤ 0.001. ns non-significant.

wounds[19,20], although full-thickness wounds in *Lepr^db/db* will eventually heal without intervention. Nevertheless, this is one of the best mouse models of impaired wound healing and the use of mice further facilitates genetic manipulation. Therefore, we sought to evaluate the actual impact of IL-1R1 signalling in diabetic wounds, by producing a *Lepr^db/db* mouse deficient for IL-1R1 (*Lepr^db/db-Il1r1^−/−*). Using this system, we revealed that diabetic mice deficient for IL-1R1 heal wounds significantly faster compared to diabetic mice that have IL-1R1. This observation strongly supports that IL-1R1 signalling is a key factor that prevents wound closure in the diabetic condition and it prompted us to explore IL-1R1 inhibition through the delivery of IL-1Ra as a possible treatment for chronic wounds.

Efficient delivery of biologics such as growth factors and cytokines in chronic wounds and more generally in regenerative medicine applications is challenging, due to their low stability and

a short window of activity following treatment[16,41]. These issues can be solved by delivering much higher doses, however, high doses of growth factors and cytokines may trigger serious adverse effects[42]. In addition, the use of high doses of therapeutics is likely to make the therapy less cost-effective and, therefore, less scalable. Recombinant IL-1Ra (anakinra, Kineret) is approved for the treatment of rheumatoid arthritis and neonatal-onset multisystem inflammatory disease, while it has been used off-label for a variety of dermatologic conditions and type 2 diabetes[13,43,44]. However, IL-1Ra is usually used at very high doses (more than 100 mg per injection) with multiple administrations and its usage can lead to side effects such as immunogenicity and infections[13,14]. Similarly, the use of high doses of recombinant PDGF-BB (becaplermin, Regranex) has been approved to treat chronic wounds, but the product raised major concerns regarding safety and cost-effectiveness[16,45,46]. Therefore, better delivery systems need to be developed to ensure precise localisation and retention of low doses of these drugs at the delivery sites. One of the strategies is to engineer recombinant factors to strongly bind a biomaterial carrier or the endogenous ECM of the tissue where they are delivered[41]. We have previously shown that engineering growth factors to bear the ECM-binding sequence of PlGF confers super-affinity to ECM components[15]. Such a strategy where the therapeutic possesses its own "built-in" delivery system has the advantage of working without the need for biomaterials, thus potentially facilitating clinical translation and reducing costs. Here, we used super-affinity IL-1Ra and PDGF-BB in a similar fashion, to allow retention in the wound and gradual release via proteases that cleave the ECM-binding sequence[15]. Although, we have previously reported PlGF₁₂₃₋₁₄₄ as the ECM-binding sequence in PlGF, we found here that a slightly shorter sequence, PlGF₁₂₃₋₁₄₁, binds ECM components with somehow higher affinity. Therefore, we utilised PlGF₁₂₃₋₁₄₁ to design super-affinity IL-1Ra and PDGF-BB and showed that both engineered factors can be retained in an ECM-mimetic hydrogel and in skin tissue for an extended period of time.

To evaluate the capacity of super-affinity IL-1Ra to promote wound healing, we compared the engineered antagonist to super-affinity PDGF-BB, since topical delivery of wild-type PDGF-BB is recognised to be one of the most promising strategies to promote healing of chronic wounds[47]. A previous study reported that extremely high doses (supraphysiological) of wild-type IL-1Ra (750 µg of anakinra) delivered with a gelatin-transglutaminase gel accelerated wound closure in diabetic mice[48]. Here, we found that wound treatment with a more therapeutically relevant dose of wild-type IL-1Ra (three orders of magnitude lower, 0.5 µg) delivered only once, and without a biomaterial carrier has only a marginal effect on wound closure in diabetic mice. However, the same lower dose of super-affinity IL-1Ra was able to significantly accelerate healing with wounds nearly closed 1 week post treatment. One issue that could arise when delivering IL-1Ra is the risk of dampening the immune response against bacterial infection, however, the use of antiseptics and systemic antibiotics are common in chronic wound management[49]. Furthermore, low doses of super-affinity IL-1Ra should not have a systemic effect, because of its ability to localise only at the delivery site, as observed with other PlGF-fused proteins[50].

Interestingly, we found that treatment with super-affinity IL-1Ra was slightly faster at promoting wound closure compared to a similar dose of super-affinity PDGF-BB, indicating that delivering immunomodulators may be more important and beneficial than delivering morphogens. Nevertheless, inhibiting the IL-1 pathway with super-affinity IL-1Ra or targeting the PDGF-BB pathway with super-affinity PDGF-BB resulted in a similar outcome, when looking at the later time point post treatment. The mechanisms by which both proteins promote wound healing in diabetic mice

are probably distinct. In the case of super-affinity PDGF-BB, the growth factor likely stimulates wound healing by multiple pathways. For instance, PDGF-BB regulates the proliferation and migration of various mesenchymal lineage cell types that are important for wound healing[16]. In addition, PDGF-BB is well-known to have a critical role in promoting angiogenesis and stabilising new blood vessels[16,51]. To understand the underlying mechanisms by which super-affinity IL-1Ra promotes healing of diabetics wounds, we first focused on wound neutrophils and macrophages, because elevated number of these cells coupled with immune dysregulation contributes to the development of non-healing wounds[2]. For instance, an abnormally high number of neutrophils in chronic wounds leads to an over-production of pro-inflammatory cytokines, reactive oxygen species (ROS), extracellular traps and proteases (MMPs and serine proteases)[52]. Similarly, failure to convert wound macrophages to an anti-inflammatory phenotype leads to high levels of pro-inflammatory cytokines (e.g. IL-1β and IL-6) and proteases, as well as reduction of key growth factors such as VEGF-A, PDGFs and TGF-β1[53]. Altogether, this creates a non-healing microenvironment where wounds cannot progress into the proliferation and resolution phases, due to exacerbation of inflammatory signals combined with ECM degradation and low levels of morphogenetic factors. Remarkably, we found that treatment with super-affinity IL-1Ra reduces the percentage of neutrophils but increases the percentage of macrophages amongst cells harvested from the wounds. Neutrophil and macrophage percentage calculations are influenced by the cell type content present in the wound (i.e. increased or decreased numbers of other immune cells and non-immune cells). However, the differences we have observed are probably not the result of cell type content variations, because neutrophil and macrophage percentages did not follow the same direction. Importantly, macrophage polarisation towards an anti-inflammatory phenotype was likely more advanced in IL-1Ra/PlGF$_{123-141}$ treated wounds, because macrophage surface expression of CD206 was higher. Further supporting the lower levels of inflammatory immune cells following treatment with super-affinity IL-1Ra, the wound concentration of pro-inflammatory cytokines (IL-1β, IL-6, CXCL1) and MMPs (MMP-2, MMP-9) were significantly lower, while anti-inflammatory cytokines (TGF-β1, IL-4, IL-10) and TIMP-1 concentration were higher. While IL-1 has been shown to promote an angiogenic response in various contexts through VEGF-A secretion by myeloid cells[54,55], the role of IL-1 in neo-angiogenesis during wound healing is still elusive. For instance, IL-1R1–deficient mice do not show impaired angiogenesis in oral wounds[56] and VEGF-A expression is decreased in skin wounds of IL-1Ra–deficient mice[57]. In this study, we found that blocking IL-1 signalling with super-affinity IL-1Ra leads to higher concentrations of wound healing growth factors (FGF-2, PDGF-BB, VEGF-A), likely promoting angiogenesis and wound closure. Together, the pro-repair microenvironment supported by super-affinity IL-1Ra treatment can be possibly attributed to a higher number of growth factor-expressing M2-like macrophages[53] in the wound, increased expression of anti-inflammatory cytokines, as well as lower levels of MMPs which degrade ECM components and growth factors[16,28–31].

Finally, we explored the effect of the treatment with super-affinity IL-1Ra on wound fibroblast senescence, because it is a major hurdle in diabetic wound healing[34–37]. Mechanistically, it has been reported that overproduction of ROS by immune cells at the wound site causes ECM and cell membrane damage which results in premature cell senescence[58]. In addition, inflammatory macrophages have been shown to activate a senescence programme in dermal fibroblasts[53,59]. Senescent dermal fibroblasts contribute to wound chronicity by secreting elevated levels of

pro-inflammatory cytokines, chemokines and proteases[60]. Here, we found that IL-1β likely contributes to dermal fibroblast senescence, because stimulation with the cytokine increases β-gal activity and induces the secretion of pro-inflammatory cytokines and chemokines which are typically associated with a senescence phenotype[60]. Further supporting that IL-1β drives senescence, wound treatment with super-affinity IL-1Ra was able to reduce the apparent dermal fibroblast senescence detected at 9 days post treatment to levels observed in non-injured skin. Together, our results suggest that the delivery of super-affinity IL-1Ra reduces dermal fibroblast senescence to near baseline levels by directly inhibiting IL-1R1 signalling in fibroblasts and possibly indirectly through decreasing IL-1β-producing neutrophil and inflammatory macrophage fractions in the wound.

In conclusion, we highlight that IL-1R1 signalling is a major factor that prevents the healing of diabetic wounds. Simple inhibition of IL-1R1 by topical delivery of an engineered matrix-binding form of IL-1Ra which allows precise localisation and retention in tissue without the need of a biomaterial carrier promotes rapid wound closure. Wound treatment with super-affinity IL-1Ra re-establishes a pro-healing microenvironment which is characterised by lower levels of pro-inflammatory cells, cytokines and senescent fibroblasts and higher levels of anti-inflammatory cytokines and growth factors. Overall, super-affinity IL-1Ra holds potential for clinical translation in the treatment of chronic wounds and may also be used in other inflammatory conditions where IL-1R1 signalling plays a pathologic role.

## Methods

**Mice.** Wild-type C57BL/6 mice were obtained from the Monash Animal Research Platform. BKS.Cg-Dock7$^m$ +/+ Lepr$^{db}$/J (Lepr$^{db/db}$) mice were obtained from the Jackson Laboratory. Because Lepr$^{db/db}$ mice are sterile, Lepr$^{db/+}$ mice were crossed to Il1r1$^{−/−}$ mice[21] to obtain fertile Lepr$^{db/+}$-Il1r1$^{−/+}$ mice. Then, Lepr$^{db/+}$-Il1r1$^{−/+}$ were crossed together to obtain Lepr$^{db/db}$-Il1r1$^{−/−}$ mice. Animals were kept under specific pathogen-free conditions. All animal experiments were conducted in accordance with Monash University guidelines and approved by the Monash Animal Research Platform ethics committee (#13395 approved 18/08/2017, #17075 approved 28/05/2019).

**Macrophage isolation and stimulation.** Bone marrow from femora and tibiae of C57BL/6 mice (8-week old) was flushed out with Dulbecco's Modified Eagle Medium/Nutrient Mixture (DMEM/F12 medium, Gibco) using a 27-gauge needle and a syringe. Cells were filtered through a 70 μm nylon strainer, centrifuged at 500 g for 10 min at 4 °C and resuspended in DMEM/F12 medium containing 10% heat-inactivated FBS, 100 mg/ml penicillin/streptomycin and 20 ng/ml murine M-CSF (R&D Systems). Cells were plated in 150 mm diameter Petri dishes at a density of $5 \times 10^7$ and cultured for 7 days at 37 °C and 5% CO$_2$. Medium was replaced every 3 days. After 7 days, macrophages were detached using TrypLE (Gibco) containing 3 mM EDTA and seeded in 12-well plates at a density of $2 \times 10^5$ cells per well in DMEM/F12 with 10% heat-inactivated FBS and 100 mg/ml penicillin/streptomycin. For measurement of secreted IL-1β, macrophages were unstimulated or stimulated with 100 ng/ml LPS (InvivoGen) for 3 h followed by 5 mM ATP (InvivoGen) for 21 h. The concentration of IL-1 released in the media was measure by ELISA (IL-1 DuoSet ELISA kit, R&D Systems). For macrophage stimulation with IL-1β, cells were co-stimulated with IL-1β (1 ng/ml) and IL-1Ra variants at increasing concentrations (0 to 1 μg/ml of IL-1Ra or equimolar concentration of IL-1Ra/PlGF$_{123-141}$). After 24 h, the concentration of IL-6 released in the media was measured by ELISA (IL-6 DuoSet ELISA kit, R&D Systems).

**Blood glucose measurements.** Non-fasted blood glucose levels were measured in 10-week-old Lepr$^{db/db}$, Lepr$^{db/db}$-Il1r1$^{−/−}$ and Lepr$^{db/+}$. Briefly, tail veins were pricked and small blood samples (2–5 μl) were collected followed by measurement of blood glucose concentration using a FreeStyle Optium Neo metre (Abbott). Any values exceeding the maximum readout of the glucometer were recorded at 500 mg/dl.

**Skin wound healing model.** The backs of male mice (12–14-week old) were shaved and two to four full-thickness punch-biopsy wounds (5 mm in diameter) were created as described previously[20]. The wounds were treated topically with 10 μl saline (PBS) or protein solution in PBS (0.5 μg IL-1Ra, 0.61 μg IL-1Ra/PlGF$_{123-141}$, 0.5 μg PDGF-BB, 0.65 μg PDGF-BB/PlGF$_{123-141}$). The wounds were covered with a non-adhering dressing (Adaptic, Johnson & Johnson) and adhesive film dressing (Hydrofilm, Hartmann). For wild-type mice, the backs of 10–12-

week-old male mice (C57BL/6) were shaved and two 5 mm diameter full-thickness biopsy punches were created on the upper middle of the back. The wounds were then splinted with 6 mm nylon rings (Zenith, ITW Proline) and the rings were covered with small adhesive circular bandages (Elastoplast, Beiersdorf). The rings were then further covered with one layer of adhesive surgical tape (Blenderm, 3 M) going around the mouse. The following day, wounds were treated topically with 10 μl saline (PBS) or IL-1Ra/PlGF$_{123-141}$ (0.61 μg in PBS). Bandages were removed and re-applied every other day. At specific time points post wounding, animals were humanely euthanised via $CO_2$ inhalation and the wounds were harvested for biochemical or histological analysis.

**Histological analysis.** Wounds were harvested with an 8 mm tissue biopsy punch and fixed in 10% neutral buffered formalin for 24 h at room temperature. The samples were trimmed until the edge of the wound, embedded in paraffin and serially sectioned into 4 μm slides until the centre of the wound was passed. The extent of re-epithelialization was measured by histomorphometric analysis. Slides were stained with hematoxylin and eosin and wound centres were determined by measuring the distance between the panniculus carnosus muscle gap using Aperio ImageScope Viewer (Germany). The extent of wound closure is calculated as the ratio between the epidermis closure over the length of the panniculus carnosus gap.

**Biochemical analysis of the wounds.** An area of 8 mm in diameter was excised from skin tissue with a wound in the centre of each biopsy, minced with scissors and incubated for 1 h at room temperature in T-PER Tissue Protein Extraction Reagent (500 μl per wound, Thermo Fisher Scientific) containing one tablet of protease inhibitor for 10 ml (Roche). Samples were then centrifuged at 5000 g for 5 min and supernatants were stored at −80 °C. Total protein concentration was measured with a Bradford assay (Millipore). Cytokines, growth factors and MMPs and TIMP-1 were detected by ELISA from R&D Systems; Mouse IL-1ra/IL-1F3 DuoSet ELISA ELISA; Mouse IL-1 beta/IL-1F2 DuoSet ELISA; Mouse IL-6 DuoSet ELISA; Mouse CXCL1/KC DuoSet ELISA; Mouse FGF basic/FGF2/bFGF DuoSet ELISA; Mouse/Rat PDGF-BB DuoSet ELISA; Mouse VEGF DuoSet ELISA; Mouse TGF-beta 1 DuoSet ELISA; Mouse IL-10 Quantikine ELISA Kit; Mouse IL-4 DuoSet ELISA; Total MMP-2 Quantikine ELISA Kit; Mouse Total MMP-9 DuoSet ELISA; Mouse TIMP-1 DuoSet ELISA.

**Recombinant PlGF fragment production and purification.** PlGF fragment sequences were cloned into the expression vector pGEX6P-1 (GE Healthcare). A histidine tag (6x histidine) was added at the C terminus of fragments. The fragments were expressed in E. coli BL21 (DE3). Bacteria were cultured overnight in 10 ml lysogeny broth (LB) medium with 100 μg/ml of ampicillin overnight. Then the culture was diluted 1:100 in 250 ml of LB medium with 100 μg/ml of ampicillin and cultured at 37 °C for 3 h. Protein production was induced with 1 mM of isopropyl β-D-1-thiogalactopyranoside (IPTG) overnight at 25 °C. Then, the culture was centrifuged at 4000 g for 10 min. The pellets were resuspended in cold PBS with one tablet of protease inhibitor cocktail (Roche), 50 mg of lysozyme (Roche). The solution was sonicated for 20 s with maximum amplitude for 3–4 cycles. Benzonase (500 U, Millipore), 1 mM $MgCl_2$ and 1% Triton X-100 were added and the solution was incubated on a rotor for 30 min at 4 °C. Lysate was centrifuged at 12,000 g for 10 min and the supernatant filtered through a 0.22 μm filter. Proteins were first purified using a GSTrap HP 5 ml and secondly with a HisTrap HP 5 ml (GE Healthcare) affinity columns. Chaperone proteins were removed by using an ATP buffer (50 mM Tris-HCl, 150 mM NaCl, 10 mM $MgSO_4$, 2 mM ATP, pH 7.4). A Triton X-114 buffer (PBS with 0.1% Triton X-114) was used to remove lipopolysaccharides. The final protein solution was dialysed against PBS and filtered through a 0.22 μm filter. The fragments were verified as >99% pure by SDS-PAGE and stored at −80 °C.

**Binding of PlGF fragments to ECM proteins.** ELISA plates (96-Well Medium Binding, Greiner bio-one) were coated with solutions of 100 nM of human plasma fibronectin (Sigma), human vitronectin (Peprotech), human tenascin C (R&D Systems) or human fibrinogen (Enzyme Research Laboratories) in PBS for 1 h at 37 °C. Wells were washed with washing buffer (PBS-T, PBS with 0.05% Tween-20) and blocked with 1% BSA in PBS-T for 1 h at room temperature. Then, wells were incubated with 100 nM of GST-fused PlGF fragments (in PBS-T with 0.1% BSA) for 1 h at room temperature. GST (100 nM) only was used as a negative control. Wells were washed three times with washing buffer and incubated with 0.1 μg/ml of HRP-conjugated antibody against GST (GE Healthcare, RPN1236V) in PBS-T with 0.1% BSA for 45 min at room temperature. Wells were then washed three times with PBS-T and detection was done with tetramethylbenzidine substrate and measurement of the absorbance at 450 nm.

**Binding of PlGF fragments to heparan sulfate.** Heparin-binding plates (Corning, #354676) were coated with 25 μg/ml of heparan sulfate (Sigma-Aldrich, H7640) overnight at room temperature and blocked with a PBS solution containing 0.2% gelatin and 0.5% BSA for 1 h at room temperature. Then, plates were washed three times with washing buffer (100 mM NaCl, 50 mM NaAc, 0.2% Tween-20, pH 7.2). GST-fused PlGF fragments (100 nM in PBS with 0.5% BSA) were added and incubated for 1 h at room temperature. Plates were washed three times with the

washing buffer and bound GST fragments were detected with HRP-conjugated antibody against GST (0.1 μg/ml in PBS-T with 0.1% BSA; GE Healthcare, RPN1236V). Wells were washed three times with PBS-T and detection was done with tetramethylbenzidine substrate and measurement of the absorbance at 450 nm.

**Recombinant protein production and purification.** IL-1Ra/PlGF$_{123-141}$ and PDGF-BB/PlGF$_{123-141}$ were designed to have a 6x histidine tag at their N terminus and the PlGF$_{123-141}$ sequence at the C terminus. Recombinant proteins were produced in E. coli BL21 (DE3) via pET-22b (Novagen) as previously described[22]. Bacteria were cultured overnight in 10 ml LB medium with 100 μg/ml of ampicillin overnight. Then the culture was diluted 1:100 in 1 l of LB medium with 100 μg/ml of ampicillin and incubated at 37 °C for 3 h. Protein production was induced with 1 mM of IPTG overnight at 25 °C. Then, the culture was centrifuged at 4000 g for 10 min. Following protein production and bacterial lysis, the soluble fraction of IL-1Ra/PlGF$_{123-141}$ was purified by affinity chromatography using a chelating SFF(Ni) column with an extensive Triton X-114 wash (0.1% v/v) to remove lipopolysaccharides. PDGF-BB/PlGF$_{123-141}$ was extracted from inclusion bodies using a solubilisation buffer (50 mM Tris, 6 M GuHCl, 10 mM DTT, pH 8.5). The extracted proteins were then added drop by drop in a refolding buffer (50 mM Tris, 1 mM GSH, 0.1 mM GSSG, pH 8.2) at 4 °C, over 4 days, for a final protein solution to buffer ratio of 1:100. The refolded proteins were then purified by affinity chromatography using a chelating SFF(Ni) column with an extensive Triton X-114 wash (0.1% v/v) to remove lipopolysaccharides. The fraction containing dimers were pulled together. IL-1Ra/PlGF$_{123-141}$ was stored in PBS with 5 mM EDTA while PDGF-BB/PlGF-2$_{123-141}$ was stored in 4 mM HCl. Murine wild-type IL-1Ra and PDGF-BB were purchased from Peprotech.

**Binding-affinity of PlGF$_{123-141}$-fused IL-1Ra and PDGF-BB for ECM proteins.** ELISA plates (Medium binding, Greiner Bio-one) were coated with 100 nM of ECM proteins in 50 μl of PBS for 1 h at 37 °C; human plasma fibronectin (Sigma), human vitronectin (Peprotech), human tenascin C (R&D Systems), human fibrinogen (Enzyme Research Laboratories). Then, wells were washed with PBS-T (PBS with 0.05% Tween-20) and blocked with 300 μl PBS-T containing 1% BSA for 1 h at room temperature. ECM and control wells (no ECM and blocked with BSA) were further incubated 1 h at room temperature with solutions of murine PDGF-BB (Peprotech), IL-1Ra (R&D Systems) or PlGF$_{123-141}$-fused proteins at concentrations ranging from 0 to 100 nM (50 μl in PBS-T containing 0.1% BSA; PROKEEP tubes, Watson bio lab). Then, wells were washed three times with PBS-T and bound PDGF-BB and IL-1Ra variants were detected using biotinylated antibodies in PBS-T containing 0.1% BSA. Antibodies used were from PDGF-BB DuoSet ELISA (R&D Systems, DY8464) for PDGF-BB and IL-1ra/IL-1F3 DuoSet ELISA (R&D Systems, DY480) for IL-1Ra. To calculate the dissociation constants ($K_d$) specific-binding values were fitted with a one site-specific binding model using $A_{450}$ nm = $B_{max}$ * [growth factors or IL-1Ra variants]/($K_d$ + [growth factors or IL-1Ra variants]).

**Dermal fibroblast isolation.** C57BL/6J mouse was humanely euthanised by $CO_2$ overdose and tail was cut off from the base. An incision was made along the midline, throughout the length of the tail from the base to the tail tip and the skin was peeled off from the bone. Tails were rinsed in PBS, cut into 1–2 cm$^2$ pieces and incubated with 2 mg/ml ice-cold dispase II (Sigma) at 4 °C for 10–12 h. Then, skin pieces were washed, and the epidermis along with hair follicles was peeled off. The dermal pieces were minced and digested in with 300 U/ml collagenase II (Sigma) for 30 min at 37 °C. The supernatant was collected and filtered with a 100 μm filter and inactivated with EDTA (5 mM). Cells were centrifuged at 500 g for 10 min and plated on T75 flasks (one flask per tail) in fibroblast media (DMEM with 2 mM GlutaMAX, 1 mM sodium pyruvate, 10% heat-inactivated FBS, 100 units/ml penicillin and 100 mg/ml streptomycin). All cells were used within the first 2–3 passages for experiments.

**Proliferation assay.** Dermal fibroblasts (passage 4) were starved for 24 h in low-serum α-MEM (100 mg/ml penicillin/streptomycin, 2 mM GlutaMax, 2% FBS). Then, cells were seeded in a 96-well plate (2000 cells/well) with low-serum α-MEM containing PDGF-BB variants for 72 h. Percentage increase in cell number was calculated over basal proliferation (low-serum α-MEM only) as previously described[61] using CyQuant Cell Proliferation Assay (Thermo Fisher Scientific). The equation used was ((cell number in stimulation group/cell number in basal proliferation group) − 1) × 100.

**Release assay from ECM-mimetic hydrogel.** ECM-mimetic hydrogels (50 μl) were generated from a HEPES solution (20 mM, 150 nM NaCl, pH 7.4) containing 8 mg/ml human fibrinogen (Enzyme Research Laboratories), 1 mg/ml human plasma fibronectin (Sigma), 500 μg/ml human vitronectin (Peprotech), 50 μg/ml human tenascin C (R&D Systems), 50 μg/ml heparan sulfate (Sigma) and 500 ng/ml of PDGF-BB or IL-1Ra variants. Matrices were polymerised in Ultra-Low Cluster 96-well plate (Corning) at 37 °C for 2 h with 10 U/ml bovine thrombin (Sigma) and 5 mM $CaCl_2$. Then, matrices were transferred to Ultra-Low Cluster 24-well plate (Corning) containing 500 μl of buffer (20 mM Tris-HCl, 150 mM NaCl, 0.1% BSA, pH 7.4). Control wells that served as 100% released control contained only PDGF-BB and IL-1Ra variants in 500 μl of the buffer. Every 24 h,

buffers were removed from wells and kept at −20 °C. Wells were replenished with fresh release buffer. For the 100% release control well, 20 μl of buffer was taken out every day and stored at −20 °C. After 7 days, the cumulative release of PDGF-BB and IL-1Ra variants was quantified by ELISA using the 100% released control as reference (PDGF-BB DuoSet, IL-1Ra/IL-1F3 DuoSet; R&D Systems). For release assays with plasmin, the same method was used except that the release buffer contained 100 μU/ml of plasmin (Roche).

**Intradermal retention assay.** Back of C56BL/6 mice (8-week old) were shaved and 10 μl of 6x histidine-tagged wild-type (IL-1Ra-His, Sapphire Biosciences) or IL-1Ra/PlGF$_{123–141}$ in PBS were injected intra-dermally. Injection sites were marked, and mice were euthanised directly after (100% control) or after 1, 3, 5 and 8 days. Full-thickness skin tissue was harvested, and the area of the injection site was collected with a 6 mm biopsy punch. Tissue was transferred into 500 μl of T-PER Tissue Protein Extraction Reagent (Thermo Fisher Scientific) containing a protease inhibitor cocktail (one tablet for 50 ml, Roche) and minced. Samples were incubated for 1 h at room temperature under agitation and centrifuged at 5000 g for 5 min and supernatants were stored at −80 °C. The concentration of IL-1Ra-His or IL-1Ra/PlGF$_{123–141}$ was determined by ELISA utilising an anti-histidine tag capture antibody (Abcam, ab18184) and a detection antibody from IL-1Ra/IL-1F3 and PDGF-BB DuoSet ELISA kit (R&D Systems). The percent retention was calculated using the day 0 concentrations as the 100%.

**In vivo phenotyping of neutrophils and macrophages.** Mice (12-week old) were wounded as described above and euthanised after 6, 9 and 14 days. The back skin was excised and wounds were harvested with an 8 mm biopsy punch and placed into RPMI media 1040 (Gibco) and 2% BSA. Wounds were minced with scissors and digested with collagenase XI for 30 min at 37 °C. Enzymatic digestion was neutralised with 5 mM EDTA and the mixture was passed through a 70 μm cell strainer. Single-cell suspensions were stained for 20 min on ice using antibodies from Biolegend (2 μg/ml): BV711 anti-F4/80 (clone BM8), biotin anti-CD11b (clone M1/70), PE-Cy7 anti-CD206 (clone C0862C), BV421 anti-Ly6G (clone 1A8). UV379-streptavidin was from BD Biosciences. Fixable live/dead viability staining was done using Zombie Aqua Fixable Viability Kit (Biolegend). All flow cytometry was performed on a BD LSRFortessa X20 flow cytometer and analysed using FlowJo (BD Biosciences).

**Measurement of surface electrical capacitance.** Measurement of surface electrical capacitance was used to evaluate the integrity of the skin barrier 9 days post treatment with saline or IL-1Ra/PlGF$_{123–141}$. The back skin of non-wounded and shaved Lepr$^{db/db}$ mice were used for baseline measurements. Mice were lightly anesthetised with isoflurane and measurements were performed using a dermal phase metre (NOVA DPM 9003, NOVA Technology Corporation, Portsmouth, NH, USA). The dl0 = instantaneous reading mode was used and the device probe was placed on the wound area. The final readings were given in dermal phase metre arbitrary units (a.u.), ranging from 90–999.

**Immunostaining of angiogenesis.** Immunostaining was performed on paraffin sections with a sodium citrate (10 mM sodium citrate, 0.05% Tween 20, pH 6) antigen retrieval step. Slides were then washed with 0.01 M PBS containing 0.05% Tween 20 (PBS-T) for 5 mins followed by a permeabilisation step in 0.01 M PBS containing 0.2% Triton X-100 for 4 min at room temperature (RT). Sections were washed with PBS-T and blocked with 1% bovine serum albumin (BSA) with 10% normal goat serum in 0.01 M PBS for 2 h at RT. For primary antibody immunostaining, sections were incubated with rabbit anti-CD31 (0.8 μg/ml Abcam, ab124432) and mouse anti-desmin (1 μg/ml, Abcam, ab8470) in 0.1% BSA with 1% NGS in 0.01 M PBS (antibody buffer) overnight at 4 °C. Sections were washed with PBS-T and incubated with biotinylated goat anti-rabbit IgG (1 μg/ml, Thermo Fisher Scientific, B2770) in antibody buffer for 2 h at RT followed by streptavidin Alexa Fluor 594 (2 μg/ml, Life Technologies, S11227) and goat anti-Mouse IgG Alexa Fluor 488 (2 μg/ml, abcam, ab150117) in antibody buffer for 2 h at RT. Stained sections were washed with PBS-T then nuclear counterstained with 4′,6-Diamidino-2-phenylindole dihydrochloride (DAPI) (1 μg/ml, Sigma Aldrich D9542-10MG) for 15 min at RT and mounted with Fluoroshield (Sigma Aldrich, F6182). Measurement was performed by taking images of three different sections from the middle of the wound area covering the dermis and epidermis (Zeiss Z1 Inverted Fluorescence Microscope) at a 20x magnification. The area positive for CD31 was measured using ImageJ software (National Institutes of Health, USA).

**Senescence-associated β-galactosidase activity.** Mice (12-week old) were wounded and treated as described above. Mice were humanely euthanised by CO$_2$ overdose and injured skin was harvested 9 days post treatment using an 8 mm biopsy, Healthy skin tissue was collected from an uninjured area of the dorsal skin. Samples were minced and digested with 1.5 mg/ml Collagenase XI (Sigma), at 37 °C for 45 min twice. Collagenase was inactivated with 5 mM EDTA and the supernatant was passed through a 70 μm filter. Cells were first stained with LIVE/DEAD Fixable Aqua Dead Cell Stain Kit (1:500 diluted in PBS, Invitrogen) for 30 min at 4 °C, followed by incubation with 10 μg/ml TruStain FcX anti-CD16/32 (clone 93, BioLegend) antibody diluted in flow cytometry buffer (PBS with 5% FBS and 5 mM EDTA). Then, cells

were stained with BV711 conjugated Biolegend rat anti-mouse antibodies against CD11b (2 μg/ml, clone M1/70), F4/80 (5 μg/ml, clone BM8) and CD45 (1 μg/ml, clone 30-F11) diluted in flow cytometry buffer, for 30 min at 4 °C. SA-β-gal staining was performed using CellEvent Senescence Green Flow Cytometry Assay Kit (Invitrogen, #C10840), according to the manufacturer's instructions. Then, cells were washed with PBS containing 1% BSA and permeabilised with 0.25% Triton X-100 in 1% BSA in PBS for 15 min at room temperature. Cells were washed with flow cytometry buffer containing 0.25% saponin, and further stained with Abcam rabbit monoclonal antibodies against Vimentin (0.2 μg/ml, clone EPR3776), Hsp47 (25 μg/ml, clone EPR4217) and S100A4 (7.5 μg/ml, clone EPR14639(2)) for 45 min at 4 °C, followed by staining with secondary antibody Alexa Fluor 647–conjugated goat anti-rabbit IgG (Abcam, #ab150083) for 45 min at 4 °C. Finally, cells were washed and resuspended in flow cytometry buffer for analysis on BD LSRFortessa X-20. The post acquisition analysis was done using FlowJo software (BD Biosciences). For the in vitro experiment, fibroblasts (passage 2) were cultured in 10% FBS with either IL-1β (1 ng/ml) or PBS control. Media was changed twice at days 3 and 6. At day 9, cells were stained with LIVE/DEAD Fixable Aqua Dead Cell Stain Kit and CellEvent Senescence Green Flow Cytometry Assay Kit.

**Statistics and reproducibility.** All data are presented as means ± SEM. Statistical analyses were performed using GraphPad Prism 8 statistical software. Significant differences were calculated with Student's t test or by analysis of variance (ANOVA), followed by the Bonferroni post hoc test when performing multiple comparisons between groups. $P < 0.05$ was considered as a statistically significant difference. The symbols *, ** and *** indicate P values less than 0.05, 0.01 and 0.001, respectively; n.s., not significant.

**Reporting summary.** Further information on research design is available in the Nature Research Reporting Summary linked to this article.

## Data availability
The authors declare that the data supporting the findings of this study are available within the article and its supplementary information files. The source data underlying the graphs of the main and supplementary figures are available in Supplementary Data 1.

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

## Acknowledgements

This work was partially funded by a Whitaker International Fellowship to B.L., the Swiss National Science Foundation (P2ELP3_175071) to Z.J., the Australian Research Council (DE170100398) to M.M.M. and the National Health and Medical Research Council (APP1140229 and APP1176213) to M.M.M. The Australian Regenerative Medicine Institute is supported by grants from the State Government of Victoria and the Australian Government.

## Author contributions

J.L.T., B.L., R.K., B.N., Y.-Z.L., C.P., Z.J. and M.M.M. conducted the experiments and analysed the data. M.M.M., J.L.T. and B.L. wrote the manuscript. M.M.M. supervised and designed the research.

## Competing interests

M.M.M. is inventor on U.S. Patent 9,879,062 which covers one of the technologies reported in this article. Monash University has filed for patent protection on the molecular design described herein, and Z.J. and M.M.M. are named as inventors. The other authors declare no competing interests.
