## [Peer Review File · Communications Biology]

Reviewers' comments:

Reviewer #1 (Remarks to the Author):

This study elucidates the role of IL-1 in dermal wound healing and presents a strategy to promote regeneration dermal lesion using a ECM binding IL-1R antagonist. Using a diabetic mouse model, they first validate the hypothesis that IL-1 signaling is an upstream regulator of the inflammatory status of dermal wounds in diabetic mice. Using, a IL-1R1 ko mouse model they further show that dermal wound healing is accelerated in absence of IL-1R signaling. Based on these findings they propose a strategy of local delivery of IL-1Ra possessing a high affinity to ECM proteins as way of altering the pro-inflammatory status of chronic dermal wounds and promoting regenerative processes, more importantly re-epithelialization. This was accomplished by fusing IL-1Ra recombinant protein with an ECM-binding sequence derived from the heparin-binding domain of placenta growth factor (PIGF123-152). This recombinant fusion protein was shown to be biological active while exhibiting strong affinity to ECM moieties such as fibronectin, vitronectin, tenascin, fibrinogen and HS. By incorporating this recombinant fusion protein in an ECM-mimetic hydrogel they authors demonstrate the therapeutic potential of the delivery strategy. Overall, this is a very well-designed study, with conclusions that are supported by the data at hand. The paper is well written, concise and statistical treatment of the data is appropriate.

There are a few points that the authors need to address:

1. The barrier properties of the re-epithelialized wound needs to be demonstrated. For example, one could assess the topic permeability using fluorescent labelled dextran.
2. While, the authors do a great job in showing the efficacy of the recombinant IL-1Ra fusion protein in promoting wound healing in a punch biopsy model, chronic wounds are a consequence of sustained inflammation leading to an environment hostile to regenerative processes. A punch biopsy model is considered a non-healing defect, that is it does not spontaneously heal. It's not clear it can be considered a chronic wound model? There are well established full-thickness skin models in pigs that are considered to etiologically similar diabetic ulcers. One could consider the ischemic rabbit ear models and also the ischemic flap models in pigs. (see Disease Models & Mechanisms (2014) 7, 1205-1213 doi:10.1242/dmm.016782). The authors need to provide a clear rationale for their choice of the animal model. The wound healing in diabetic mice is not considered to be similar to that observed in human subjects.
3. The isolation of macrophages using adherence assay is a bit problematic as adherence leads to changes in macrophage phenotype. Why did the authors not choose to isolate peripheral blood monocytes and differentiate them to macrophage lineage?
4. Under dermal fibroblast isolation: Please clarify and include euthanasia procedure before dermal fibroblast isolation
5. Please include institutional review board approval info such as protocol number, approval date etc)
6. Regarding the biomimetic-ECM. A reference (peer-reviewed literature) to this ECM-mimetic hydrogel should be included.

Few minor points:

Line 132. should be "...activity of engineered.."

I would suggest that the results and discussion be combined to reduce redundancy, unless the journal format necessitates a separate section.

Reviewer #2 (Remarks to the Author):

This manuscript provides new information about the effective use of a novel bioengineered molecule to improve diabetic wound healing. The findings are supported by excellent data presentation, and the approach is novel. The work will be of broad interest to those who study the basic biology of wound healing, chronic wounds, poorly healing wounds, and growth factor delivery. A few minor suggestions are made to improve the manuscript.

1) The sentence in the abstract that states "Generating a knockout mouse model, we demonstrate that IL-1R1 delays wound closure in diabetic conditions" is difficult to understand and seems an odd way to state the conclusion. The knockout mouse shows that the elimination of IL-1R leads to improved healing, but to say that IL-1R itself delays healing seems a bit odd, as more commonly the ligand would be the factor that is considered to mediate the delay. Indeed, the results state the situation quite clearly, as they suggest that the increase in IL-1 β and decrease in IL-1Ra that delay healing in the condition of diabetes.

2) The PDGF-BB/PIGF construct seems to improve diabetic healing quite effectively, although perhaps not quite as well as the IL-1Ra/PIGF construct. The authors might want to discuss how targeting disparate pathways (IL-1 vs. PDGF) could both lead to similar outcomes. Does the administration of PDGF lower IL-1 levels in this or other studies, or are they independent pathways of improvement?

3) A limitation of the flow cytometric analysis shown in Figure 3F is that the data depicts percentages of total cells rather than actual numbers present at the wound site. If the treated wounds are more cellular (due to increased fibroblast and/or endothelial cell content), that could skew the calculation, making it appear for example, that less neutrophils are at the site. The relative number of total cells and total inflammatory cells that are present in treated and untreated wounds should be included and considered in the report to assist in understanding the data.

Reviewer #3 (Remarks to the Author):

The ms entitled "Restoration of the healing microenvironment in chronic wounds with matrix-binding IL-1 receptor antagonist" by Jean L. Tan et al., shows that a chimeric IL-1Ra-PIGF(122-143) protein binds the ECM, and therefore enables slower kinetics and longer effect of IL-1R inhibition, allowing the use of lower levels of IL-1Ra concentration to be used. This effect results in better wound healing response in the diabetic Lepr db/db mouse model, by inhibiting pro-inflammatory response and promoting M2 phenotype wound healing response.

While the idea that IL-1Ra is important for wound healing by increasing TGF- β signaling pathway was first published by Ishida et al., in a 2006 JI paper entitled "Absence of IL-1 Receptor Antagonist Impaired Wound Healing along with Aberrant NF- κ B Activation and a Reciprocal Suppression of TGF- β Signal Pathway", the idea of administering IL-1Ra in a more practical manner than using Anakinra is novel and interesting.

It looks as the authors decided to take the basic idea shown in Ishida et al., paper to a applicative project in which the administration of the therapeutic factor is important not less than the factor itself. The idea behind the experiments are well described, the experiments are described and written in well, and the technical data looks solid.

During the review of the MS several points were raised:

1) I would be keen to see better description of the wound healing process, that is a more detailed

photographs of the wounds on the back of the mice or more detailed histology, to those that were treated with PIGF-IL-1Ra vs IL-1Ra or saline (Figure 3b).

2) I believe that a protein composed of PIGF peptide fused to a non-relevant protein can better strengthen the results, since both therapeutic proteins the PIGF-IL-1Ra and the PIGF-PDGF contain the same PIGF peptide. Therefore, a protein that binds the wound area in high affinity as control can be informative.

3) In addition, or alternatively to point 2, the Lepr db/db-IL-1R1KO can be used to show whether the PIGF-IL-1Ra protein has any effect. Although this mouse has better response than Lepr db/db, in this mouse the IL-1Ra part of the protein has no significance.

4) How this PIGF-IL-1Ra treatment affect wound healing response in non-diabetic mice is an interesting question. If the authors have this kind of data, it would be of an interest to show.

5) Carmi et al "The Role of Macrophage-Derived IL-1 in Induction and Maintenance of Angiogenesis" paper among others have shown that IL-1 induce pro-angiogenic response. Blocking IL-1 and angiogenesis should be explained and discussed in the MS.

Reviewer #1:

This study elucidates the role of IL-1 in dermal wound healing and presents a strategy to promote regeneration dermal lesion using a ECM binding IL-1R antagonist. Using a diabetic mouse model, they first validate the hypothesis that IL-1 signaling is an upstream regulator of the inflammatory status of dermal wounds in diabetic mice. Using, a IL-1R1 ko mouse model they further show that dermal wound healing is accelerated in absence of IL-1R signaling. Based on these findings they propose a strategy of local delivery of IL-1Ra possessing a high affinity to ECM proteins as way of altering the pro-inflammatory status of chronic dermal wounds and promoting regenerative processes, more importantly re-epithelialization. This was accomplished by fusing IL-1Ra recombinant protein with an ECM-binding sequence derived from the heparin-binding domain of placenta growth factor (PIGF123-152). This recombinant fusion protein was shown to be biological active while exhibiting strong affinity to ECM moieties such as fibronectin, vitronectin, tenascin, fibrinogen and HS. By incorporating this recombinant fusion protein in an ECM-mimetic hydrogel they authors demonstrate the therapeutic potential of the delivery strategy. Overall, this is a very well-designed study, with conclusions that are supported by the data at hand. The paper is well written, concise and statistical treatment of the data is appropriate.

There are a few points that the authors need to address:

1. The barrier properties of the re-epithelialized wound needs to be demonstrated. For example, one could assess the topic permeability using fluorescent labelled dextran.

Response: In the revision, we have assessed the barrier properties of the re-epithelialized wounds by repeating some of the wound healing experiments and by measuring the skin surface electrical capacitance with a Dermal Phase Meter. The surface electrical capacitance directly correlates with the permeability of the epithelium (Goretsky, M.J. et al., Sander, E.A. et al.). Because of time constraints and feasibility, we have decided to compare the two most important groups, the saline control and IL-1Ra/PIGF. The new data (Supplementary Fig. 5b) show that wounds treated with saline have high capacitance values, while wounds treated with IL-1Ra/PIGF₁₂₃₋₁₄ display values that are similar to those measured on uninjured skin. This indicates low fluid leakage and therefore reformation of an epithelial barrier.

- Goretsky, M.J., Supp, A.P., Greenhalgh, D.G., Warden, G.D. & Boyce, S.T. Surface electrical capacitance as an index of epidermal barrier properties of composite skin substitutes and skin autografts. *Wound Repair Regen* 3, 419-425 (1995).
- Sander, E.A., Lynch, K.A. & Boyce, S.T. Development of the mechanical properties of engineered skin substitutes after grafting to full-thickness wounds. *J Biomech Eng* 136, 051008 (2014).

2. While, the authors do a great job in showing the efficacy of the recombinant IL-1Ra fusion protein in promoting wound healing in a punch biopsy model, chronic wounds are a consequence of sustained inflammation leading to an environment hostile to regenerative processes. A punch biopsy model is considered a non-healing defect, that is it does not spontaneously heal. It's not clear it can be considered a chronic wound model? There are well established full-thickness skin models in pigs that are considered to etiologically similar diabetic ulcers. One could consider the ischemic rabbit ear models and also the ischemic flap models in pigs. (see Disease Models &

Mechanisms (2014) 7, 1205-1213 doi:10.1242/dmm.016782). The authors need to provide a clear rationale for their choice of the animal model. The wound healing in diabetic mice is not considered to be similar to that observed in human subjects.

Response: We agree with the reviewer. In the first version of the manuscript, we made sure to not claim that the *db/db* mouse model is a model of chronic wounds. Indeed, this model is considered as an impaired wound healing model and only mimics some aspects of chronic wounds in humans (stated in the result section, page 4). We now also clarify the rationale of using this model in the discussion section (page 15, paragraph 2).

3. The isolation of macrophages using adherence assay is a bit problematic as adherence leads to changes in macrophage phenotype. Why did the authors not choose to isolate peripheral blood monocytes and differentiate them to macrophage lineage?

Response: Isolation of peripheral blood monocytes and differentiation into macrophages could have been used. However, we believe that one would also need to plate the blood monocytes in a cell culture dish (make them adhere to the plate) with M-CSF to differentiate them into macrophages, which is similar to what we did with bone marrow-derived cells. Moreover, the vast majority of blood monocytes in the mouse originates from bone marrow and are CCR2⁺/Ly6c^{high} (the small fraction of non-classic Ly6c^{low} monocytes is considered as vascular resident). Monocytes enter into the blood circulation, flowing inflammation/injury, via the CCR2-CCL2 axis (Kimball, A. *et al.*, Teh, Y.C. *et al.*). Overall, while adherence to a plate can lead to changes in macrophage phenotype, we think that this cannot be avoided with the use of either bone marrow-derived or blood-derived monocytes. We now explain the reasoning of using bone marrow-derived monocytes in the discussion section (page 15, paragraph 1).

- Kimball, A., *et al.* Ly6C(Hi) Blood Monocyte/Macrophage Drive Chronic Inflammation and Impair Wound Healing in Diabetes Mellitus. *Arterioscler Thromb Vasc Biol* 38, 1102-1114 (2018).
- Teh, Y.C., Ding, J.L., Ng, L.G. & Chong, S.Z. Capturing the Fantastic Voyage of Monocytes Through Time and Space. *Front Immunol* 10, 834 (2019).

4. Under dermal fibroblast isolation: Please clarify and include euthanasia procedure before dermal fibroblast isolation.

Response: We now include the euthanasia procedure (page 29).

5. Please include institutional review board approval info such as protocol number, approval date etc).

Response: We now include the ethics ID number and the approval dates (page 20).

6. Regarding the biomimetic-ECM. A reference (peer-reviewed literature) to this ECM-mimetic hydrogel should be included.

Response: It is the first time we report this biomimetic-ECM. The ECM-mimetic hydrogel is based on a fibrin matrix functionalized with fibronectin, vitronectin, tenascin C, and heparan sulfate. All ECM components are naturally incorporated in the hydrogel thanks to protein-protein interactions, following thrombin-induced fibrin polymerization. Fibronectin is crosslinked to fibrin via the transglutaminase factor XIIIa (present in fibrinogen preparations) (Corbett, S.A. et al.). Vitronectin directly interacts with fibrin (Podor, T.J. et al.), while tenascin C binds fibronectin (Chung, C.Y. et al.). Fibrin, fibronectin, vitronectin, and tenascin C all have heparin-binding domains capturing heparan sulfate in the hydrogel. We now give a better description of the biomimetic-ECM in the result section (page 7, paragraph 3).

- Corbett, S.A., Lee, L., Wilson, C.L. & Schwarzbauer, J.E. Covalent cross-linking of fibronectin to fibrin is required for maximal cell adhesion to a fibronectin-fibrin matrix. *J Biol Chem* 272, 24999-25005 (1997).
- Podor, T.J., et al. Incorporation of vitronectin into fibrin clots. Evidence for a binding interaction between vitronectin and gamma A/gamma' fibrinogen. *J Biol Chem* 277, 7520-7528 (2002).
- Chung, C.Y., Zardi, L. & Erickson, H.P. Binding of tenascin-C to soluble fibronectin and matrix fibrils. *J Biol Chem* 270, 29012-29017 (1995).

Few minor points:

Line 132. should be "...activity of engineered.."

Response: We have changed the sentence accordingly.

I would suggest that the results and discussion be combined to reduce redundancy, unless the journal format necessitates a separate section.

Response: We have kept the original structure but removed some redundancies in the discussion section.

Reviewer #2:

This manuscript provides new information about the effective use of a novel bioengineered molecule to improve diabetic wound healing. The findings are supported by excellent data presentation, and the approach is novel. The work will be of broad interest to those who study the basic biology of wound healing, chronic wounds, poorly healing wounds, and growth factor delivery. A few minor suggestions are made to improve the manuscript.

1) The sentence in the abstract that states “Generating a knockout mouse model, we demonstrate that IL-1R1 delays wound closure in diabetic conditions” is difficult to understand and seems an odd way to state the conclusion. The knockout mouse shows that the elimination of IL-1R leads to improved healing, but to say that IL-1R1 itself delays healing seems a bit odd, as more commonly the ligand would be the factor that is considered to mediate the delay. Indeed, the results state the situation quite clearly, as they suggest that the increase in IL-1beta and decrease in IL-1Ra that delay healing in the condition of diabetes.

Response: We agree with the reviewer and now states “Generating a knockout mouse model, we demonstrate that the IL-1–IL-1R1 axis delays wound closure in diabetic conditions.” to be precise and avoid confusion.

2) The PDGF-BB/PIGF construct seems to improve diabetic healing quite effectively, although perhaps not quite as well as the IL-1Ra/PIGF construct. The authors might want to discuss how targeting disparate pathways (IL-1 vs. PDGF) could both lead to similar outcomes. Does the administration of PDGF lower IL-1 levels in this or other studies, or are they independent pathways of improvement?

Response: We now discuss this point in the discussion section (page 17, paragraph 2). The mechanisms by which IL-1Ra and PDGF-BB promote wound healing in diabetic mice are likely distinct. Administration of PDGF-BB probably does not directly lower IL-1 levels, while the growth factor may have some immunomodulatory activity (either positive or negative) on immune cells.

3) A limitation of the flow cytometric analysis shown in Figure 3F is that the data depicts percentages of total cells rather than actual numbers present at the wound site. If the treated wounds are more cellular (due to increased fibroblast and/or endothelial cell content), that could skew the calculation, making it appear for example, that less neutrophils are at the site. The relative number of total cells and total inflammatory cells that are present in treated and untreated wounds should be included and considered in the report to assist in understanding the data.

Response: We agree with the reviewer that reporting the actual total number of immune cells per wound would provide a more accurate picture of the wound microenvironment. However, in order to calculate the absolute numbers, we would have needed to use counting beads or run the entire wound sample (something we did not do, due to the very high number of cells per wound). Because of time constraints and restrictions, we have not been able to repeat the entire flow cytometry experiment with all groups and time points. Nevertheless, we think that the percentage of neutrophils and macrophages are still a good indication of the effect of the treatments and, in our case, the calculations were perhaps not skewed. If percentage calculations were skewed by an increased number of fibroblasts and/or endothelial cells (or any other cell types in the wounds), one would

have seen the same trend in both neutrophils and macrophages. However, we found that wounds treated with IL-1Ra/PIGF have less neutrophils but more macrophages, at day 9 post-treatment. In addition, we also have reported the CD206 median fluorescent intensity which is independent of the percentage or cell number. The data indicates that macrophage polarisation towards a “M2-like” phenotype is higher in wounds treated with IL-1Ra/PIGF. Further supporting the lower levels of inflammatory immune cells following treatment with IL-1Ra/PIGF, the wound concentration of pro-inflammatory cytokines (IL-1 β , IL-6, CXCL1) and MMPs (MMP-2, MMP-9) were significantly lower, while anti-inflammatory cytokines (TGF- β 1, IL-4, IL-10) and TIMP-1 concentration were higher. We now further discuss our data and the limitation of reporting percentages in the discussion section (page 18, paragraph 1).

Reviewer #3:

The ms entitled "Restoration of the healing microenvironment in chronic wounds with matrix-binding IL-1 receptor antagonist" by Jean L. Tan et al., shows that a chimeric IL-1Ra-PIGF(122-143) protein binds the ECM, and therefore enables slower kinetics and longer effect of IL-1R inhibition, allowing the use of lower levels of IL-1Ra concentration to be used. This effect results in better wound healing response in the diabetic Lepr db/db mouse model, by inhibiting pro-inflammatory response and promoting M2 phenotype wound healing response.

While the idea that IL-1Ra is important for wound healing by increasing TGF- β signaling pathway was first published by Ishida et al., in a 2006 JI paper entitled "Absence of IL-1 Receptor Antagonist Impaired Wound Healing along with Aberrant NF- κ B Activation and a Reciprocal Suppression of TGF- β Signal Pathway", the idea of administrating IL-1Ra in a more practical manner than using Anakinra is novel and interesting. It looks as the authors decided to take the basic idea shown in Ishida et al., paper to a applicative project in which the administration of the therapeutic factor is important not less than the factor itself. The idea behind the experiments are well described, the experiments are described and written in well, and the technical data looks solid.

During the review of the MS several points were raised:

1) I would be keen to see better description of the wound healing process, that is a more detailed photographs of the wounds on the back of the mice or more detailed histology, to those that were treated with PIGF-IL-1Ra vs IL-1Ra or saline (Figure 3b).

Response: We initially did not take pictures of the back of the mice, because we think that macroscopical wound evaluation in mice is often unreliable. However, as requested, we now provide more detailed histology for IL-1Ra/PIGF, IL-1Ra and saline in Supplementary Fig. S4a. We also give a better description of the wound histology sections in the captions.

2) I believe that a protein composed of PIGF peptide fused to a non-relevant protein can better strengthen the results, since both therapeutic proteins the PIGF-IL-1Ra and the PIGF-PDGF contain the same PIGF peptide. Therefore, a protein that binds the wound area in high affinity as control can be informative.

Response: To verify if the PIGF₁₂₃₋₁₄₁ sequence itself has an influence on wound healing, we tested the effect of treating wounds with PIGF₁₂₃₋₁₄₁ fused to the non-relevant protein GST. Delivering GST/PIGF₁₂₃₋₁₄₁ had no significant effect on wound healing (Supplementary Fig. 4c,d).

3) In addition, or alternatively to point 2, the Lepr db/db-IL-1R1KO can be used to show whether the PIGF-IL-1Ra protein has any effect. Although this mouse has better response than Lepr db/db, in this mouse the IL-1Ra part of the protein has no significance.

Response: Because of time constraints, we have not been able to do this experiment (*db/db-IL-1R1 KO* are very difficult to breed, because mice homozygote for *Lepr^{db}* are sterile). However, we think that we have answered

the question of the reviewer with the GST/PlGF₁₂₃₋₁₄₁ experiment above. We have demonstrated that the PlGF₁₂₃₋₁₄₁ sequence itself has no effect on wound healing.

4) How this PlGF-IL-1Ra treatment affect wound healing response in non-diabetic mice is an interesting question. If the authors have this kind of data, it would be of an interest to show.

Response: This is an interesting point raised by the reviewer. Thus, we have tested whether delivering IL-1Ra/PlGF affects wound closure in wild-type mice. Because wild-type mice close wound very fast by contraction (an effect which is much less pronounced in *db/db* mice), we have used a model of splinted wounds. Compared to treatment with saline control, treatment with IL-1Ra/PlGF₁₂₃₋₁₄₁ led to a significant but modest improvement of wound closure (Supplementary Fig. 6a,b), suggesting that blocking IL-1R1 signalling in animals without immune dysregulation has likely less impact on wound healing.

5) Carmi *et al* "The Role of Macrophage-Derived IL-1 in Induction and Maintenance of Angiogenesis" paper among others have shown that IL-1 induce pro-angiogenic response. Blocking IL-1 and angiogenesis should be explained and discussed in the MS.

Response: We agree that it is an important point to cover. While IL-1 has been shown to promote an angiogenic response in various contexts through VEGF-A secretion by myeloid cells (Fahey, E. & Doyle, S.L.; Carmi, Y. *et al*), the role of IL-1 in neo-angiogenesis during wound healing is still elusive. For instance, IL-1R1-deficient mice do not have impaired angiogenesis in wounds (Graves, D.T.) and VEGF-A expression is decreased in wounds of IL-1Ra-deficient mice (Ishida, Y.). We now further discuss the role of IL-1 in angiogenesis and the effect we have observed when blocking it (page 18).

- Fahey, E. & Doyle, S.L. IL-1 Family Cytokine Regulation of Vascular Permeability and Angiogenesis. *Front Immunol* 10, 1426 (2019).
- Carmi, Y., *et al*. The role of macrophage-derived IL-1 in induction and maintenance of angiogenesis. *J Immunol* 183, 4705-4714 (2009).
- Graves, D.T., *et al*. IL-1 plays a critical role in oral, but not dermal, wound healing. *J Immunol* 167, 5316-5320 (2001).
- Ishida, Y., Kondo, T., Kimura, A., Matsushima, K. & Mukaida, N. Absence of IL-1 receptor antagonist impaired wound healing along with aberrant NF-kappaB activation and a reciprocal suppression of TGF-beta signal pathway. *J Immunol* 176, 5598-5606 (2006).

REVIEWERS' COMMENTS:

Reviewer #1 (Remarks to the Author):

Dear Dr. Trompouki,

The authors have addressed my concerns and I believe the manuscript is suitable for acceptance.

However, the authors need to address one point before this manuscript is accepted. The use of the term "chronic wounds" in the title is misleading. As, concurred by the authors the db/db mouse model is considered to a model for impaired wound healing. Since, there are models for chronic wound healing and these have not been used here the authors need to amend the title and I strongly believe that "impaired wounds" is more appropriate.

Kind regards,

Prasad Shastri

Reviewer #2 (Remarks to the Author):

The response to reviewer comments is quite complete and all major concerns are adequately addressed.

Reviewer #3 (Remarks to the Author):

The revised version of the MS by Dr. Martino describes the activity of the ECM-binding PIGF-IL-1Ra in wound healing process in a well-written manner.

As a reviewer, I can conclude that the data is solid, well written, the idea of fusing anti-inflammatory cytokine to the ECM, and to improve the PK and by this to achieve better healing process is of an interest.